# Carbon and Cellulose-Based Nanoparticle-Reinforced Polymer Nanocomposites: A Critical Review

**DOI:** 10.3390/nano13111803

**Published:** 2023-06-05

**Authors:** Gopal Yuvaraj, Manickam Ramesh, Lakshminarasimhan Rajeshkumar

**Affiliations:** 1Department of Mechanical Engineering, RVS College of Engineering and Technology, Coimbatore 641402, India; 2Department of Mechanical Engineering, KIT-Kalaignarkarunanidhi Institute of Technology, Coimbatore 641402, India; 3Department of Mechanical Engineering, KPR Institute of Engineering and Technology, Coimbatore 641407, India

**Keywords:** carbon, cellulose nanoparticles, polymer composites, processing methods, properties, characterization

## Abstract

Nanomaterials are currently used for different applications in several fields. Bringing the measurements of a material down to nanoscale size makes vital contributions to the improvement of the characteristics of materials. The polymer composites acquire various properties when added to nanoparticles, increasing characteristics such as bonding strength, physical property, fire retardance, energy storage capacity, etc. The objective of this review was to validate the major functionality of the carbon and cellulose-based nanoparticle-filled polymer nanocomposites (PNC), which include fabricating procedures, fundamental structural properties, characterization, morphological properties, and their applications. Subsequently, this review includes arrangement of nanoparticles, their influence, and the factors necessary to attain the required size, shape, and properties of the PNCs.

## 1. Introduction

Nanoparticles are important materials, even though they are of nanometer size. As a result of their nanoscale size range, the inter-relationships between various constituents is accomplished. The nano-composite materials are categorized into different materials based on the size of the nanoparticle. The relationship between the volume and surface area of the material also plays a vital role in the property of the materials due to its dimensional characteristics [1,2]. Cellulose nanomaterials (CNMs) have a major role in the field of current nanomaterials and are an area of increased interest for researchers due to their degradability, bio-compatibility, and economic benefits. Polymer-based nanomaterials are considered to be bio-degradable and bio-compatible, and are alternatives for several applications due to their intriguing properties. Nanoparticle widths ranging from 1 to 100 nm have novel properties, such as a huge surface-to-volume proportion, a smaller size, the capacity to embody different medications, etc. [3,4]. 

Nanocomposites fabricated along with metallic, non-metallic, and polymer-based materials provide extraordinary benefits, such as surface deformities and other characteristics of fabricated materials. These composites are characterized by their qualities that include a larger surface area, significant mechanical properties, wear and scratch resistance, and improved optical characteristics [5,6]. Carbon-based fillers in PNCs are termed as 1D straight elements such as carbon nanotubes (CNTs), 2D layered elements such as montmorillonite, and 3D powder elements such as silver nanoparticles (AgNPs). Similarly, cellulose-based nanoparticles are extracted from natural fibers. Generally, polysaccharide-based nanoparticles are classified as starch-based, cellulose-based, protein-based, and glycogen-based nanoparticles. Due to the bonding of the polymer matrix, with nanofiller a high impact in nanocomposites could be noticed. Subsequently, increasing the number of nanofillers with sizes under 100 nm may bring enhancements in the composite strength. The fabrication of nanocomposites can be undertaken by utilizing in situ strategies, the solvent technique, or by blending them with a polymer matrix. The notable characteristics of PNCs are their better thermal stability, mechanical properties, less gas porosity, etc. [7,8]. Consequently, the choice of nanoparticles is an indication of property improvement in nanocomposites [9,10]. Oliveira et al. [11] reviewed the properties of PNCs and found that they prompt a significant increase in interfacial properties when compared with conventional composite materials without nanoparticles. The interfacial region is a critical part of polymer composites that has unique characteristics in relation to the polymer, even at minimum loading conditions of the nanofiller. Interfacial construction is known to be not quite the same as mass design or for nanoparticles containing polymers with larger surface areas, despite the filler concentration. Bio-based PNCs are a type of material with moderate properties that are made from natural polymers [12]. The plant fibers can be used as reinforcing agents in the polymer composites due to advantages such as high strength, solidness, minimal effort of processing, and low thickness, and they produce low CO_2_ outflow [13,14]. The focus of this review was to highlight various aspects of carbon and cellulose-based polymer nanocomposites such as their synthesis, properties, and applications. Despite the availability of numerous studies, the current review emphasized the advanced characterization and applications of PNCs with state-of-the-art research and review works. Subsequently, the following section deals with the general aspects, synthesis methods, properties, characterization, and applications of carbon and cellulose-based PNCs. The review also highlighted the challenges involved in the full-scale implementation of the PNCs along with the future scope of these materials. 

## 2. Polymer Nanocomposites

Nanocomposites are selected based on their properties, including dimensional quantities, quality, similarity with the micro-structural lattice, and the distribution of nano-material throughout [15]. Past researchers have found that the evenly distributed oxide components in the nanoparticles improve the composites’ fire retardance, mechanical behavior, and thermal properties when compared to micro-composites [16,17]. The nanoparticles have a major role in impacting the molecular structural characteristics of the polymer matrix. They produce a higher impact on the molecular structure of the material, which directly influences the material’s behavior [18,19]. Nanoparticle scattering might be described by a very fine phase at the nano level. The accumulation of nanoparticles in a material’s surface can be explained from the morphological behavior [20]. Different methods are implemented to improve the scattering quality along with one or another substance of the actual material [21]. 

The distribution of the nanoparticles on the surface and along the depth of the material was studied through morphology and optical property analysis [22,23]. The surface treatment is done to help the similarity of the structures and fillers utilized, with the help of the intersection of the organo-silanes or long-chain platelets’, intercalating particles, etc. [24]. An alternate uniform scattering should be performed to attain a high degree of nanocomposite properties [25]. Recent investigations of CNT with polymers have suggested that the catalysts are metallic nano-sized particles such as iron, nickel, and cobalt, which can be sprayed on silicon substrates either by solution spraying, electron beam evaporation, or physical sputtering. The layer thickness depends on various factors such as particle size, the deposition process, and the ability to control the particle size, which is relevant for the development of nanocomposites [26,27] based upon composition, and their microstructures are shown in Figure 1a. The enhancement in the mechanical, thermal, electrical, and rheological properties of PNCs relies on various variables, such as fabrication procedure, interfacial relation among nanoparticles and polymers, and the condition of nanoparticle distribution [28]. The fabrication process can affect the bonding strength and modify the dimensions of the nanoparticles, affect surface penetration, decrease the combustibility, or increase the mechanical properties based upon the volume-to-surface ratio, as shown in Figure 1b [29].

Coatings, in turn, enhance the impact resistance of nano-sized fillers—at the periphery of the substrates whilst also requiring a decreased quantity of nanoparticles than the mass [30]. The basic requirement for increasing the mechanical properties of PNCs with nanoparticles is a lack of distributions within the polymer chain [31,32]. Nano-coatings are probably applied by using electro-spraying, which can follow layer arrangements, such as substance or chemical vapor deposition (CVD), which enable the application of a layer on a nano scale [33]. Electro-spraying is typically used to determine gas pressure and is incorporated in constant formation lines [34]. Nano-coatings enable surface functionalization and explicit properties such as anti-microbial, self-recuperating, fire retardance, gas obstruction, etc. [35]. In the case of bundling, packaging applications are probably multiplied through the coatings of nanocomposites just as a fuel line boundary, anti-microbial activities, and so on [36]. The most used nanofillers are organo-clay bentonite, quasi 1D graphite, etc. [37,38]. Additionally, coatings are probably applied for tweaking surface proclivity in the direction of fluids and gels [37,39] to obtain non-stick repellent sheets that can be easily removed [40,41,42]. Mechanical strengthening confers the nanoparticles an inverse kind of strength and stability, and permits them an enhanced capacity at a decreased weight compared with composites or pure polymers [42,43,44,45,46]. Phases of non-obligatory electricity contribute to the various utilization and depths of connections, both within the solar reacted sheets in securing layers and at in outdoor applications [47,48]. 

Power generation through 2D structural graphite is intended within the outer region [49,50]. Nano-textured surfaces and nano-coatings are created to supply self-cleansing influences and to restrict the electricity yield [51,52]. Coatings of nano-structured substances have been applied on the outer layer to enhance stickiness and boundary layer thickness [53,54]. A distinctive method that can be applied for nanocomposite transformation is increasing the moist compound in addition to electro-spraying [55,56]. A substance can add carbon functionalities to fibers at the nano scale, for example, arrangement, scattering of the filler, and interfacial bonding among the filler and polymer [57,58]. The obstacles and required method enhancements are then referenced, demonstrating problems in nanoparticle scattering [59]. The method often involves the elevated exploitation of nanocomposites inspected at the interfacial region, as well as mechanical, electrical/microwave characteristics, and combustibility obstruction [60,61]. PNCs are improved along with polymer volatile matters, which scatter inside the composites [62,63]. The applications of nano-composites include bundling, signboards, and other diverse uses [64,65,66]. Quantum dots such as metal particles, metal oxides, and semiconductors are major research topics in recent history due to their good optical, magnetic, catalytic, and thermal properties. The structure of quantum dots is presented in Figure 2a.

Nanotubes are similar in design to the circular atom C60 found in the 1980s; however, they are stretched to resemble rounded constructions of 1–2 nm in width [67,68,69]. In their structure, nanotubes involve solitary layers of carbon atoms arranged similarly to a chamber [67,69]. These are called single-walled CNTs (SWCNTs), which are shown in Figure 2b. They can likewise be shaped as various concentric cylinders with a measurement of 20 nm, and if they feature a length greater than 1 mm, they are called multi-walled CNTs (MWCNTs) [70,71,72]. They show high thermal and electrical conductivity, high surface territory, and conceivably high atomic adsorption capacity [73,74]. Nano-wires can be generated from conducting (metals) or semi-conducting (carbon) materials using several processing techniques [74]. They have a uniform crystal structure and diameter in the range of a few tens of nm and a high aspect ratio [75,76]. These nano-wires are used as interconnectors for the transport of electrons in nano-electronic devices. Different metals such as cobalt, gold, and copper have been used to manufacture nano-wires. Phyllo-silicates are the most widely used layered silicates for making polymer-layered silicate (PLS) nanocomposites. Layers of two tetrahedrally coordinated silicon atoms are fused into an edge-shared octahedral sheet of either aluminum or magnesium hydroxide to make up their crystal structure [77,78]. Three different forms of PLS nanocomposites are thermodynamically possible depending on the frequency of interfacial interactions between the polymer matrix and layered silicate.

**Figure 2 nanomaterials-13-01803-f002:**
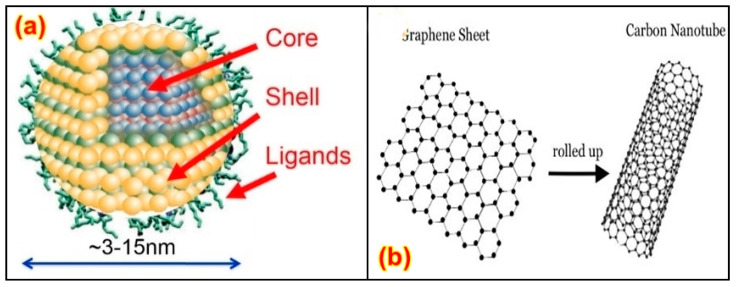
Structure of (**a**) quantum dot [66]. and (**b**) SWCNTs [68]. Reproduced from Ref. [68] with permission from Elsevier.

The graphite comprises layered nano-sheets of graphene-like layered silicates [79]. Graphite is generally utilized as a nanofiller to create conductive polymer-based composites. Traditional fillers such as metal powder, carbon dark, or carbon fibers are generally used in the micrometer scale. A filler proportion of 15–55% in polymer nanocomposites results in enhanced electrical conductivity, but at the same time, it reduces mechanical properties and increases the density of the polymer nanocomposites [80,81,82]. Incredibly, the surface roughness of the graphite nano-sheets enables the arrangement of layers inside the polymer. Graphite nano-platelets (GNPs) comprise graphite particles intercalated with prominently focused structures, which have the potential to expand 200–300-fold while maintaining the same crystallinity degree at elevated temperatures [83,84]. Isolating a graphite sheet into nano-platelets, a few graphene layers, with a high angle proportion is considered as having barely any layered graphene. Unlike other nanoparticles, graphite has special features, with a totally unreasonable modulus close to that of graphene. An arrangement of graphene–graphite layers with C-C bonding possesses higher thermal and electrical conductivity in its plane owing to the laminated structure of the model [80,85]. 

Due to the nanoscale impacts and profoundly actuated debris of the PNCs, a massive part of the coatings of nanofillers have high morphological properties with electro-chemical, active coupling, functional CNT-influenced polymers, and so on [86,87,88,89,90]. These PNCs can be utilized for material reinforcement, as electrolytes, drugs, biomedical materials, thin films, etc. [89]. The major processing methods for PNCs are sol–gel amalgamation, oxidative polymerization, thermally atomized methods, microwave-assisted processes, composite blending, electrochemical deposition, reactant chain polymerization, etc. [91,92,93]. 

### 2.1. Processing Methods

#### 2.1.1. Wet Chemical Processing

Wet chemical processing is one of the most important processing methods for use in the formation of films, which are prepared through the casting method and also for coating on unique substrates. A polymer-based coating is applied to a substrate using a spraying technique [38]. Standing out from ousted coverings, critically, these coatings have much lower impediment resistance [94]. In contrast with multilayer films, nano-coatings have reduced reinforcement usage [95,96,97], which is both a monetary and biological advantage, and the thermoplastic stages are explained in Figure 3a. They can be utilized for the reinforcement of fluid-repellent houses in different layers, e.g., for water-resistant-based bundling [2] clean-to-discharge [5,87] processes.

#### 2.1.2. Thermo-Plastic Processing

Thermoplastic processing is used for the processing PNCs, and is completed through the extrusion process, which is given in Figure 3b. Extrusion permits the quick melting of a polymer with an excessive power entry at some stage. It is possible to stuff, blend, homogenize, and plasticize the mass of the PNCs during the whole process, and they can also be modified over surface coatings [98,99]. While blending the filler materials into a modified matrix, unique properties are obtained. When processing the ordinarily preferred exfoliated nanocomposites, the dispersion particularly relies upon the extruder’s configuration [100]. Shedding upheld at maximum shear favors a higher dissipation, even for longer durations. Additionally, the spot wherein the nano-clay is added has been exhibited to be a basic segment. Regardless, the essential difficulty is whether or not excellent dissipation or stripping is viable in the thermodynamic proclivity of the nano-clay/nanoparticles and polymer structures [101,102,103].

#### 2.1.3. Physical Vapour Deposition

The advantages of liquid-phase approaches include addressing the most severe developments of layer deposition of thin layers. Depending on the reinforcing components, the vapor-phase technique may be known as physical vapor deposition (PVD) [104,105]. Nano-clay techniques are conducted in an excessive reaction and rely on the change of consistently protecting fabric on the vapor level observed through an accumulation on the surface of the substrate [106]. The mixing is carried out by using proper strategies, which can be a substance reaction amongst the fiber and matrix individually. At a vaporous length, the protecting fabric is acquired through vanishing due to electrical heating or electron beam illumination [106,107] or by using faltering strategies [108]. In the other methods, debris of the consistent protecting fiber is dispatched because of the effect of the vaporization method, which is ionized and advanced with the help of electrical charging. The boundary conditions of the saved layer are probably advanced methods for appealing difficult programming, which is called magnetron faltering [109,110]. The production of thin-walled metal films on the surface of polymeric materials with the help of PVD is generally used for steel- and polymer-blended nanocomposites [111]. 

#### 2.1.4. Chemical Vapour Deposition (CVD)

The CVD technique relies on gaseous precursors reacting chemically on the substrate surface [107,112]. The responses are enacted both with the help of using heat or with the manually graphite-formed surface, since the polymer’s primary chains were allowed to move around the lamella. This can likewise be clarified by the presence of a nano-clay structure [113,114]. Atomic layer deposition (ALD) is a modified process of a CVD method [115]. Layers form over the penetration of oxygen and water vapor through bundling films, and, in this method, blast the presence of bundled fibers [112,116]. The surfaces of vacuum protection boards, photovoltaic modules, and characteristic mellow radiating diodes, contrary to natural impacts, subsequently require liquefied intercalation at 270 °C; the matrix containing 3 and 5% organo-clay indicates the development of crossover with some inclination of peeling [117]. Electro-hydrodynamic preparation is a miniature nano-fabrication process including electro-turning and electro-spraying [102,118]. Electro-turning is a process that produces persistent polymers with measurements ordinarily within the micron range through the development of an outer layer over the top voltage-electric-controlled request constrained on a polymer’s course of action or condenses. 

Surface topography is affected by properties (fundamentally by methods addressing the consistency, surface strain, and conductivity of the polymer) and by strategies addressing their limitations [119,120,121]. With some restrictions, chemical substrates inside the layers can be acquired by changing their direction and manner, and this method is known as electro-hydrodynamic atomization [122]. The nano-clay is ordinarily made, resembling a metallic film that is one-sided by means of a high voltage. In contrast with other methods, electro-spray deposition (ESD) offers the advantage of an over-the-top execution and a decrease in the network procedure [122,123,124,125]. Nano-sponges can melt the molecule, as detailed for the meloxicam drug resembling a cyclodextrin. Meloxicam has a melting point at 250 °C. This is not the case when meloxicam is reinforced in nano-sponges, hence demonstrating the arrangement of a bond between meloxicam and the nano-sponges [96,126], and that the use of meloxicam is ideal in nano-sponges; although there have been some attempts to upgrade the electro-turning framework, it is nowadays easier than electro-splashing [79,80,81], which is scaled-up through multi-nozzle injectors to a higher level. The preparation technique is explained in Figure 4.

### 2.2. Conventional Manufacturing Techniques 

Enhancing properties emerge when the nano-structure of a material is correlated with the basic aspects of material science. More than one exposing surface can cause the organo-clay characterization to move out of different materials [81]. A few of the resulting nanocomposites may exhibit their ultimate properties, while the others may show strengths related to morphological structures [82,127]. In contact with these zones, measurements of electrical, mechanical, thermal, and optical characteristics demand examination such that the nanocomposites can be used in common applications. In the arrangement of fibers in nano-sized polymer matrix composites (PMCs), the preliminary step is the selection of the proper technique. A few of the most widely applied techniques for processing PMCs are lay-up, spraying, pultrusion, resin transfer molding (RTM), vacuum-assisted process, the autoclave method, pre-preg method, fiber winding, etc. [83]. Wet lay-up is a fundamental approach that contrasts with other nano-composite manufacturing techniques, as presented in Figure 5. The mechanical characteristics of the composites are poor owing to the presence of voids and the result is non-uniformity. Pultrusion is a process with an excessive advent rate, wherein the pre-preg or materials gather and form a cluster. Some conventional nano-composite preparation processes are presented in Table 1.

We investigated MWCNTs’ impacts on ramie fiber-reinforced epoxy composites with different weight fractions in terms of their mechanical and crack propagation characteristics. The results of this investigation demonstrated an increase of inter-laminar shear and flexural strengths along with flexural moduli by 38%, 34%, and 37%, respectively, according to the volume fraction of MWCNTs. The inclusion of CNTs increased the storage moduli, glass transition temperature, and quasi-static crack propagation. Conversely, durability was diminished by adding different concentrations of CNTs [84,85,128,129]. Higher durability is probably obtained via utilizing a woven fabric fiber. VARTM is considered to be a modified version of the RTM technique, where open molds are used to reinforce the additives by means of a vacuum [130]. This setup has advantages compared with hand lay-up, and it diminishes surface defects when compared with other methods. The impact of nano-constituents on the structure causes an issue with the flow of resin consistency and fastens energy [131], and there is probably a threat of voids over the reinforcements [132,133]. Autoclaving is a promising method for manufacturing excellent composite structures and can process thermosetting and thermo-plastic composites with uniform thicknesses and the least porosity [134]. For advanced composite materials for airplanes, autoclaving remains one of the most widely applied methodologies [135].

Resin film imbuement (RFI) is much the same as RTM, in which a thin film is reinforced on top of the layer under heat and weight. Resin-infused woven or unidirectional-oriented fibers are reinforced inside the pre-preg strategy with vacuum packing and autoclave molding. Although the technique is difficult to work with, the pre-form is ordinarily homogenous [136,137]. In the filament winding technique, infused filaments are folded around a mandrel and shaped into the required form. Muddled round and hollow segments, strain vessels, and gas and water tanks for automotive applications might be fabricated with the help of this method [138]. The consistency-related issues of the matrices of this technique might negligible [139]. Fiber winding is a cost-effective method for manufacturing round and barrel-shaped components. This is an advanced method for conventional composite processing [140,141]. 

As of now, a number of experiments are generally organized on the level of nanoparticle scattering in polymer matrices. For the most part, the effect of scattering degree on mechanical and rheological properties is perceived, which is shown in Figure 6 [142]. The non-appearance of synthetic reinforcements among the polymer and molecule, in association with scattering, can affect the mechanical properties of PNCs owing to the solid movement of the load from fiber to the matrix due to the presence of a solid interfacial bonding. Ultrasonic molding of nanocomposites was performed, and it was found that the measure of energy is disturbed at the interfaces [143,144]. As a result, this method is often used to disperse, emulsify, and trigger particles [145,146]. Furthermore, ultrasound energy can disrupt C–C bonds, which are important for the formation of long-chain radicals, which results in the formation of a chemical bond on the surface of nanocomposites [143,147]. Ultrasound techniques have been used to separate silica agglomerates in flexible ethylene propylene diene monomer (EPDM), drastically lowering the size of agglomerate particles. Menezes et al. [148] discovered that the added nano-silica powder during ultrasound processing was diminished and redispersed in fluid structures. Wang et al. [149] prepared an ultrasound strategy for polyolefin-earth nanocomposites utilizing MMT soils changed with quaternary ammonium salts in a polyolefin resin. This system typically impacts the blend of nanoparticles to address immoderate aggregation in the nanocomposites, as shown in Figure 7. 

Surfactant-assisted mechanical alloying and milling minimize clustering by providing a barrier to reduce surface strain [150,151]. Three stages can be identified in the milling process for crystalline nanoparticles in a high-power ball mill. Deformation localization within the shear bands occurs first, followed by the development of a grain shape in the nano-dimensions [152]. Hydrophobic surfaces can be obtained with the essential property of surfactants to prevent debris from accumulating and to determine the length and shape of the final product based on the surfactant grinding material and process [153,154]. Ye et al. [154] analyzed carbon’s impact resistance from 80 to 500 nm with the use of sodium dodecyl sulfate. They found that the particles displayed more grounded scattering and accumulation while poly (acrylic ruinous, sodium salt) became surfactant. Consequently, high-force ball milling can be applied for the arrangement of extraordinary properties for use in reinforcements [155].

The utilization of CNTs in polymer composites enables attractive thermal, mechanical, optical, and electrical properties. Properties of CNT such as elasticity, chirality, and so on are governed by their physico-mechanical properties [156]. The advantage of CNTs is their top-notch elastic energy and moduli, and they are foreseen as having application in reinforcements [157]. Polymer/CNT composites have attracted interest because of their specific surface areas and strong interfacial bonding with the matrix due to their nanoscale microstructure model [158]. The reinforcing components of CNTs are considered to be a particular form of these reinforcing fibers, and fillers will enhance the resulting properties [159].

A number of studies have been performed on the experimental setup arrangements of CNT composites by using polymer matrixes [160,161], and it has been found that the tensile and bending strength, modulus, and yield strength increase significantly. Among different polymers, PET is the most widely utilized polymer with different nanoparticles and fibers. The distinction between conventional carbon fibers produced using polymer and catalytically grown carbon filaments is within the diameter, and various handmade fibers can be consistent, while other fibers are irregular in shape [162,163]. Fibers additionally vary inside the microstructure of the carbon, as shown in Figure 8a [164]. They have carbon layers randomly oriented close to the fiber structure, while other fibers do not have this orientation. Due to the measurement of the carbon fiber morphology, the dispersion of carbon fibers in a polymer matrix requires additional attention when compared with that of conventional carbon fibers in composites. A technique for scattering the carbon fibers in a thermoplastic resin is slurry preparation, including fiber arrangement, mixing the slurry with nano-powder, depleting the appropriate response and drying, and, finally, heating over a glass transition temperature [165,166,167]. On account of a thermosetting polymer, for example, epoxy, the scattering of fibers induces the weakening of the interface into a dissolvable state, and this serves as an approach to bring down the consistency and resulting blending of the fiber with the matrix [168,169]. Figure 8b shows the morphology of various carbon-based structures. 

Because of the high impact of the state of the resin during the scattering of the fibers, the mechanical and electromagnetic properties of the composites fundamentally rely on the fiber texture and orientation [170]. Imoisili et al. [171] separated two types of nano-fibers: CNFs made of cone-molded carbon layer stacking and by peeling carbon fibers, and another type is the vapor-generated nanofibers [170,172]. The range is no longer the most effective in terms of morphology, despite the addition of nano-substantial values. A few works have been carried out on nano-fiber composites and their compounds, utilizing procedures including over-the-top double-screw pultrusion and low-shear in-situ polymerization and infusion forming with diffusion. CNF modification has been specifically carried out through plasma polymerization, cleaning and coating, and corrosive and plasma oxidation, which results in hydrophilicity in the fiber surfaces and the deposition of non-graphitic carbon [173]. Development from an external perspective of oxidized graphite nanoparticles through glycidol has been carried out [168,174]. The hydroxyl center was promoted by CNF methods and this carboxylic-added carbon nano-clay may be used to support dispersion of resources of CNFs or solvents [175]. Composites with oxidized CNFs have been arranged by reinforcing untreated CNFs. The scattering of oxidized CNFs exhibited has good thermal stability, shear moduli, and better glass transition temperatures, being the best situated for nanocomposites.

Graphite’s properties make it a potential polymer filler, resulting in applications in electromagnetic interference shields and thermal conductors [127]. During the manufacturing of the GNP composites, a low similarity among graphene sheets was a major problem during their manufacturing [82]; this is shown in Figure 9. Another related problem is the issue of scattering GNPs inside the polymer to increase uniform mixtures [82,127]. As with carbon fibers in the wake of assembly, graphite, in its layered structure, has little usefulness. Different modification techniques have been adopted for carbon fibers, and most of these systems can be actualized to GNP/epoxy nanocomposites [83].

During the process, plasma treatment has been utilized to change the GNP surfaces; powder volatility and plasma chamber contamination through vacuum venting and siphoning are the most prominent issues. Ultra-sonication and UV treatment of graphite powders and epoxy-based nanocomposites containing GNPs improve the forming and scattering of graphite into the matrix [81]. Permeable multi-layer graphite has enabled superior electrical conductivity and reduced mechanical characteristics when compared with pristine graphite [82,83], which is described to enhance modifications in graphite surface and morphology. The inclusion of modified GNP in epoxy resin transformed the direction of loading into the opposite of that of the film’s direction. This demonstrated an advancement in the conveyance of the graphene composites in the frequency range of 400–750 nm [85].

## 3. Properties of PNCs

### 3.1. Mechanical Properties

Nanofillers are incorporated in composites to enhance their mechanical characteristics, including immovability and quality, by processing methods [133]. Specific results can be explained explained depending on the addition of treated silicon oxide in the nanoparticles, including effects on their dissipating area, poly-dispersity, organo-change, etc. [134,136]. It is particularly endeavored that well-spread and aligned nano-platelets can overhaul the immovability of the matrix material. Increasing the Young’s modulus results in imbuement-formed composites with similar moduli and a pure polymeric amide cross-section [137]. Their properties are according to the 10–15 µm diameter of the glass fibers. This outcome is noteworthy for a composite formed by using a thin film stacking or injection molding process [138,139,140]. The characteristics of polymers are influenced by the size and properties of the nano-sized particles, e.g., modulus of rigidity and thermal conductivity [141]. Besides, it was shown that even once the compound lattice advances from the cleaned stage to the polyanile stage, a lot of energy must be contributed to yield the mechanical properties and moduli which are tabulated in Table 2.

Along these lines, the component of the inter-phase layer in nanocomposites is generally used somewhere near net-shaped products. Heat-added nano-phases were determined to boost the inter-phase of compounds [144]. As considered, thermally blended depolarization loosened up the compound; a larger part of compound was then decided by the surface zone. Atomic force microscopy (AFM) is used to investigate chemical bonding along with the particle surface and inspect the outcome of the medium among nanoparticles and their compounds [143,146]. The third zone is the compound area of the coated nano-sized particle, and it is an unprecedented material [147]. Consequently, it is feasible to depict the surface first by tests, inside the resulting stage, by a common numerical model. Hence, data regarding the third stage of the inter-phase could then be eliminated. In this way, homogeneous nucleation shows up in region-disengaged blends [143,177]. Additionally, limited quantum dots of particles in a medium are valuable to nucleation for nanoparticles of 29–50 nm and at any rate of a dispersed method for 5–24 nm pores [143]. 

With larger nano-sized particles, less crystallization energy is required by virtue of the upper viscousness. This is observable in various composites, including polycarbonate, polymeric amide [167,169], polylactide/nanoclay [168], polymeric MMT [171], polymeric amide multi-walled fullerene, polyester/nanoclay, poly (butyleneterephthalate)/nanoclay, polypropylene/nanoclay, and polypropylene-multi-walled fullerene [174,175,178,179,180]. While nucleation is evident in the composites, the crystallization rate is routinely lowered. A change in nanocomposites is seen through the glass transition temperature and by adding nanoparticles [181]. 

### 3.2. Flammability Properties

Many applications in the fields of construction, road transport, and aerospace need increased fire retardancy properties [72]. Some automobile producers conjointly have explicit guidelines identifying the flammability of vehicle components [74,77]. This becomes a critical analysis for property alternatives in line with safety standards, particularly relating to halogenated compounds. Consequently, a few examinations are applied to develop new ecologically agreeable halogen flammability retardant additives or to expand their strength [76,86]. The issue emerging for some standard fire retardant mixes is that they have extraordinary requirements, which together influence the mechanical properties. The inclusion of nanofillers allows the improvement of fire retardancy [68,70,71,72,73,74] and the mechanical properties at the same time, as shown in Figure 10. The examinations were conducted with respect to the consequence of muds into different polymers, demonstrating an increment of the fire retardancy [71,74]. While nanofillers are normally insufficient to satisfy guidelines for fire retardancy, major reinforcing methodologies are usually applied for the mixing of conventional fire-retardant substances and nanofillers [182]. The effect of scattering on the combustibility nature is driven by the obstruction of component pervasion, which will stop the ignition compared with nanoparticles.

### 3.3. Characterization of PNCs

The characterization of any substances is a basic requirement for its use. Recent studies showed how materials behave in common applications [23,24]. These include the techniques required to characterize the materials, for example, steel, alloys, semi-conductors, polymers, nano-structures, etc. The characterization is dependent on mechanical, electrical, optical, and thermal properties and the synthesis of materials [25,26,27,28,29]. Morphological investigations demonstrated that the increased content of nanoparticles conferred a high agglomeration rate, and when the content was further increased, it resulted in higher agglomerations that directly affected the mechanical properties of the resulting nanocomposites. The major characterization techniques of PNCs are compiled and presented in Table 3.

#### 3.3.1. Raman Spectroscopy

Raman spectroscopy is an unprecedented technique for the characterization and isolation of ordinary or possibly inorganic blends in composite substances. As light passes through a model, a small portion of the image is dispersed. The majority of dispersed light already the same extraordinary properties as the incident light, which is known as Rayleigh scattering. Raman scattering occurs when a part of the light is dispersed at a high frequency [32]. The Raman effect is the efficiency capacity between incident light and Raman-scattered light [38,40,131]. The Raman spectrum is a plot of Raman depth versus the frequency of the Raman shift [46]. It involves sharp peaks from PNCs, which can be typical for determining the relationship between the nanoparticles and the polymer matrices [47,48]. A Raman spectrum may be obtained from tests and can be as small as 1 mm [48,49]. The powers of a spectrum in a Raman range depend upon the affectability of the specific vibrations to the Raman influence and can be useful for exploring the spectrum between cellulose-based nanoparticles. As such, Raman spectra can be utilized for intensive examinations of the spectral images of the PNCs [49,50]. 

By a wide margin, the most prominent portion of the scattered light is said to be Rayleigh scrambling [81,82]. A part of the light is scattered at a high frequency; this is named a Raman diffusion [84,129]. Raman effect signs can routinely be obtained via a solitary model; each is related to a pivotal vibration or rotational movement of PNCs. In each viable sense, since, in reality, the Raman effect is so effective, a laser is utilized as the wellspring [130]. It includes sharp social occasions which can be common in the supportive relationship of the mixes or substances [132,133]. Raman spectroscopy is a remarkable strategy for the conceptual evaluation and seclusion of standard or potentially inorganic mixes of carbon and cellulose-based PNCs [131,134]. In some studies performing PNC characterization, this approach is utilized for its ability to affirm atoms, both in mass and as character particles. 

#### 3.3.2. Scanning Electron Microscopy

Scanning electron microscopy (SEM) is used to examine the surfaces of the particles and inner structures [54,56]. This is an important technique to visualize any surface that can endure in a vacuum. The SEM images begin with a thin layer of gold coating applied to the surface of the nanofiller or nanoparticles to falter it [59,60,61,62,63,64]. In some studies, it is stated that if the surface of the PNCs is not effectively conductive, surface conductivity can be increased by coating gold nanoparticles [65,66]. This incorporates a finely collimated light emission that clears all through the outside of the examination. The pillar is centered directly into a test that filters the outside of a sample [68,182]. The transmitted particles accumulate in the proper locator to yield surface data. The assessment of the nanofiller scattering in the polymer surface is significant since the thermal and mechanical behavior of the materials is emphatically identified according to the morphology of the acquired materials. Contingent upon the level of detachment of the nanoparticles, morphologies are conceivable [69,70,71].

#### 3.3.3. Environmental SEM

It is very expensive to use modern environmental SEM (ESEM) to modify a flowing electron magnifying lens; to proceed with SEM [72,88], a magnifying instrument intended for a double feature can work well in each mode [74]. One preferred position of utilizing the ESEM is using a wet model, in which the need to make the non-conductive nanofiller conductive is dispensed with. PNC samples do need not dried and covered with carbon or gold, palladium, and accordingly their characteristics can be saved for testing or control [76,77]. In some studies, it was mentioned that dynamic examinations can be finished with the ESEM in the sodden mode; one of the hot levels might be utilized to heat carbon-based PNCs above 1500 °C and picture them for the span of each progression of the heating/cooling measure [78,86]. After a positive temperature is surpassed, over 1100 °C, inclination should be acclimated to dismiss hot electrons [185], and this can be performed without issues. The Peltier uses a heating/cooling stage, grants a working interior temperature of 20 °C above or under the surrounding temperature, and provides a blend of low temperature and high water vapor pressure, allowing for the achievement of a 100% relative humidity (RH) at the sample surface [91,92,93,101]. At 100% RH, tests are characterized by drying all throughout the imaging strategy [101,186]. At a value that is not exactly 100% RH, a wet sample continually loses water since the vacuum inside the chamber siphons it; inside the degree, it shows up as a consistent development [112,187].

#### 3.3.4. Rutherford Backscattering Theory

Rutherford Backscattering Theory impacts nuclear cores, and is the first theory to introduce the possibility of cores in particles [99,100]. It includes estimating the dispersion and quality of particles in a pillar, which subsequently reduces the close surface area of an example into iotas where the bar has been focused. With these data, deciding the nuclear mass and essential fixations versus power under the surface is possible. Rutherford Backscattering (RBS) is appropriate for deciding the centralization of prime factors [186,188]. Its affectability for the conduction of tests appropriately under the PNCs’ surface is negative [98,99,100,184,188,189]. At the point when the carbon or cellulose-based PNC samples are subjected to light emission, the greater parts of the samples are embedded into the nanocomposite and do not move away. This is because the distance across a nuclear core is approximately 1–15 m, though the dividing among cores is around 2–10 m [103,183]. A small portion of the occurrence particles go through an immediate crash with the core of the atoms inside the sample [104,105]. This impact does not just imply direct contact between the shot particle and target atom energy, but happens because of Coulombic power between cores in close proximity to each other [109,189]. Be that as it may, with the utilization of conventional carbon or cellulose-based nanocomposite materials, the communication can be demonstrated precisely as a versatile crash [78,111]. The force is estimated for molecule backscattering and the rays pass through the sample, each when a crash [113]. Hence, this relies upon a material’s ability to halt electricity [114,120]. A molecule likewise loses quality as the after-effect of the crash. In some investigations, it was stated that the proportion of energy of the shot upon crashing is known as the kinematic energy and this energy is influential in achieving better characterization of images for carbon-based PNCs [112,115,116,117]. The quantities of backscattering occasions that emerge from a given detail in an example rely on two components: the grouping of the component and the large size of its core. 

#### 3.3.5. Energy-Dispersive X-ray Spectroscopy (EDAX)

Energy-Dispersive X-ray Spectroscopy is an incredible methodology for measuring the internal structures of a sample in terms of cubic micrometers [121,123]. In energy-dispersive spectroscopy, the particles at the surface are energized by the electron beam by discharging specific frequencies of X-ray beams, which may be normal for the nuclear state of the components [102,118,119,120,121,123]. A power-dispersive identifier [124,190], a solid-state device that separates among X-ray beam energies, examines the outflow X-ray beam [126]. In most studies, the influence of the polymer matrix and cellulose-based nanofiller can be determined using EDAX analysis. The presence of various elements can be ascertained using EDAX and most studies identify the influence of the elements on the characteristics of carbon and cellulose-based PNCs.

#### 3.3.6. Transmission Electron Microscopy

TEM is similar to SEM, in which an inordinate voltage (80–200 keV), an engaged electron beam, is emitted through a narrow, consistent case, ordinarily 100–200 nm in thickness [96,125,126,190]. Electrons are subject to sound dispersion or diffraction from grid planes inside the glass-like section of the substance. Ionized X-ray beams are identified by using different indicators [79]. TEM is used in most studies to characterize the internal characteristics of carbon- and cellulose-based PNCs. 

#### 3.3.7. Auger Electron Spectroscopy

After the surface of a solid has been flooded with electrons, Auger alludes to the venture of a secondary electron [136,138]. This is a one-of-a-kind photoelectron radiation phenomenon that occurs after a reliable surface is bombarded with low-energy X-rays. The Auger electron’s electricity is determined by the chemical bonding of the elements present in carbon and cellulose-based PNCs from which it was emitted. The intensity of an Auger electron’s escape is much less than 1 nm, with metals having a low density and the conducting cloth having the highest value. Auger electron spectroscopy (AES) has a lateral resolution of about 1 mm. Auger spectroscopy characterizes the surface of substances and represents the structure of substances in this way [137,140]. This method employs low-powered electron energy beneath 5 keV to decrease the heating and decay of the surface. 

#### 3.3.8. Ion Scattering Spectroscopy

After one or more notable crashes with target iotas of the best couple of layers, a portion of the rays will be returned into the vacuum when a light emission reaches a solid surface. Estimation of the backscattered density may be used to determine the mass of these iotas [147,176]. Low-emission ion spectroscopy (LEIS) refers to fundamental energies between 100 eV and 10 keV, medium-emission ion spectroscopy (MEIS) refers to energies between 100 and 200 keV [154], and high-emission ion spectroscopy (HEIS) refers to energies between 1 and many MeV. The LEIS approach is often referred to as ion-scattering spectroscopy (ISS), whereas the HEIS technique is referred to as RBS [143,177]. LEIS is engaged as a story-explicit strategy and the spectra are usually collected using particle radiation ranging from 0.5 to 3 keV. The probability of electron movement is extremely high, even within the fundamental impact with a surface iota, due to the strong electron affinity of inert-gas ions [161]. After the collision, the largest particles are destroyed, so a locator set to investigate particles of a similar kind as the ones in the episode shaft identifies particles that had just a single crash with an objective molecule [158,159,161]. Since rays entering the solid need multiple scattering to return to the surface and exit, they are discarded. The exceptional affectability of ISS to only the highest level layer or two mono-layers dictates their proper use [160]. In many studies, ion scattering spectroscopy has been used on cellulose-based nanocomposites for exposing surfaces, thin film coatings, and attachment, just as iotas, along with the limitation of adsorbed molecules. Measurement of surface texture and the utilization of low-quality particles, are according to the failure of the inelastic materials and the balance rate, depending on particle directions [165]. Moreover, covering tops and two or three dissipating must be contemplated, and will turn into a basic apparatus [160,162].

#### 3.3.9. Secondary Ion Mass Spectroscopy

Electron spectroscopy for chemical analysis (ESCA) spectra for indistinguishable substances are hard to fix, while structures will strengthen the polymers [95,162]. Because of the smaller sampling intensity, the secondary ion mass spectroscopy (SIMS) technique is used. In a standard evaluation, the surface of the sample is assaulted by a strategy for a primary molecule at a low thickness, basically intended to confine the change of the model surface by considering enlightenment. The surface produces nanoparticles, which might be examined with the use of a mass analyzer [95,166]. The delayed consequences of explicit evaluation give the compound its structure and create layers around the surface [162]. A model is degraded and constantly attacked with electrons, splitting into pieces that can be recognized through incomplete particles of a comparable essential mass. There are two types of mass examinations: the first one is an objective that isolates between fragmentary mass evaluations and the second one is a mass appraisal that isolates huge mass value. The procedure calls for exceptionally low weight, and the parts should be eccentric [95,159,160,161,162,163,165]. Charged fragments are ejected into a vacuum; this is not recommended, as they may be unreliable. The goal relies upon the cracking of an ionized particle [162,165]. Mass reach proposes the purposeful assessments of molecule powers [95,157]. There are different methods for the ionization of typical blends: far-reaching atom attack, compound and local ionization, electron influence, deposition, etc. [158,169]. For instance, in a laser, lower-than-anticipated experimental steps include a mono-chromatic light that is utilized to pass on the energy to volatilize a demonstration from the surface [156,157]. 

#### 3.3.10. Gas Chromatography

A blend is separated into its additives using gas chromatography (GC), which are analyzed by one of the detectors [167]. In GC, the pattern is passed through an appropriated stable bed in a column at some point during vaporization. The vapor passes on using a regular scattering of a gas such as nitrogen or helium, which is heated by the evaporator [95,171]. GC insinuates the usage of strong springy or nuclear sifter segments [167,168,169]. Chromatography isolates a mix into its added substances, which are then broken down [168,171]. In GC, the example is given at some stage in the vapor fragment through appropriated stable bedding set in a section. Heating substances heat to the segment, through which the vapor is conveyed by utilizing a customary flow of a fuel which incorporates organic gases [168]. GC alludes to the utilization of solid retentive or sub-atomic strainer sections. This technique is used in studies to identify the fundamental elements responsible for the inherent characteristics of carbon, cellulose, and their polymer nanocomposites. By identifying these elements, the influencing factors for the properties of the PNCs can easily be adjudged. 

#### 3.3.11. Nuclear Magnetic Resonance

This is a method that relies on the appealing idea of a couple of isotopic materials. In reality, it is a concept relying on engaging solicitation with a continued identification of the circle energy [171]. It is assaulted with recurrent electromagnetic waves at legitimate concentrations to the engaging area. Precisely when the turning place and the signal recurrent become the same, reverberation occurs. The quality switch is described in the middle and is surveyed and portrayed [155,170]. The resounding rehash at which the quality absorption happened depends upon the substance of the model and the natural segments. The properties in an atom might be seen by nuclear magnetic resonance (NMR) [168,180], with reference spectra and fuse assortments of substances.

#### 3.3.12. Differential Scanning Calorimetry

Differential scanning calorimetry (DSC) is a thermal evaluation procedure and is utilized to check the heat flow rate with temperature changes. This is one of the best methods among the available thermal assessment systems [147,149]. These evaluations give theoretical and substance changes that recall intense and extensive structures or changes for heat limit [95,164]. In any event, the comprehensive non-appearance of dissolvable stage advances, crystallization temperature, loosening up, and contamination of the polymers achieves an adjustment in the temperature of the model. DSC structures give a broad limit range from 80 °C to 1600 °C [172,173].

#### 3.3.13. Dynamic Mechanical Analysis

This approach is usually a greater-sensitivity method for recognizing changes compared with the DSC and DTA [173]. This is a direct result of the assessment in the dynamic modulus and damping coefficient (Tan δ), the two of which trade while the glass-like shape advances to the amorphous fragment. During the process, a moderately greater substitute in the mechanical characteristics occurs in its uncommon heat ability. Dynamic mechanical analysis (DMA) is the method for estimation of glass transition temperature and particular minor region/shape adjustments of polymers [95,172]. It chooses loss modulus, storage modulus, and Tan δ as a component of temperature or time. Values are taken at regular intervals and Tan δ as the quality of substance weight. DMA can moreover be used to control the temperature limits. A weight is finished to a model and the adequacy and portion of the subsequent movement are assessed [181]. For every circumstance, the model is pushed at a repeat of the experiments. Units typically have a repeat range of 0.001 to 1000 Hz. For most indicative tests, any assessments below 0.01 Hz take an unreasonable amount of time, particularly if the information is needed as a temperature property [175,179]. Resonation habitually happens at frequencies more than 100 Hz, contingent on test robustness. These utilize a straight actuator, which is settled from the data of magnetic impact analysis to the electromagnetic waves as activated agents [94,181]. 

#### 3.3.14. Thermo-Gravimetric Analysis

This is a system used for the measurement of the heat flow rate composites which consolidates polymer matrices [103,104]. In this method, changes inside the model are assessed at the same time as the temperature is increased. Moisture content and other substances of a model can also be examined through TGA [105,117]. The robustness is set over the radiator and is thermally remote from the light. TGA equipment is set up with a smaller-than-expected heat, which may be immediately cooled [104,105]. An external radiator with a heating focal point after the effect of a platinum and rhodium blend can increase the temperature up to 1500 °C. TGA is good for temperatures above 1000 °C, 0.1 mg balance affectability, and a variable direct heating rate. The thermal capacity of TGA may likewise change from 0.1 to 200 °C/min [78,111]. In most of the studies, it was stated that the thermal stability of the polymer matrices was enhanced by the addition of nanoparticles based on carbon substrates and cellulose microconstituents. 

#### 3.3.15. Differential Thermal Analysis

The most critical temperature of differential thermal analysis (DTA) is in the abundance of 1010 °C [170]. A standard heat-up charge for DTA is between 13 and 22 °C/min [166,173], notwithstanding that more moderate articulations are possible by the use of an ordinary ideal model of 50–100 mg load [172,173]. The dissolving of a semi-crystalline or glass-like polymer showed itself as an endothermic reaction. The glass-like part of the model is related to the region beneath the height, just as it is in DSC. Since the conditioning explanations for polymers are usually unaffected by using the aggregate, DTA may be used to represent blends of polymers [170,173]. Similar polymers, including superfluous and low thickness, are discernable by DTA; however, IR proved unable to do this, and is now set up to successfully resolve such issues, proposing the thermo-gram of a mix of polytetrafluoroethylene and perfluoroalkoxy polymer [175,179]. The peak at an atmospheric temperature (19 °C) is an advancement factor for polytetrafluoroethylene and dissolving factors [94,180].

## 4. Applications of PNCs

Carbon and cellulose-based PNCs are widely used in many application fields such as energy storage, electrical and electronics fabrication, food packaging, biomedical, electromagnetic interference (EMI) shielding, and so on. In some studies, it was stated that the use of carbon-based nanoparticles such as CNTs and carbonyl powder in the polymer matrix enhanced the EMI shielding properties of the nanocomposites. When the polymer nanocomposites were manufactured with the assistance of magnetization curing, the electrical conductivity of the PNCs was enhanced and this aids in the EMI shielding applications [191]. Some other studies developed electric heating composites based on CNTs embedded in polymer matrix and the incorporation of silica gel was also examined. From the results, it was seen that silica gel incorporation enhanced the dispersion of CNTs in the polymer matrices, thus enhancing the electric heating ability and heat-induced electrical stability of the PNCs. The addition of silica gel also increased the time of cooling, which enhanced the heat-storage capability of the composites. It was concluded from the study that the silica gel addition increased the electric heating ability of the CNTs-based PNCs [192]. Cellulose-based PNCs are highly used in food packaging applications. These composites possess better biodegradability, mechanical, permeability, barrier, and optical properties owing to their nanoscale dimensions and high specific surface area [193,194,195]. Though the content of cellulose increases the mechanical properties of the cellulose nanocomposites, excessive content can lead to agglomeration. Food packaging applications demand good permeability and barrier properties, which are exceptionally high for cellulose PNCs. Hence, the use of cellulose-based polymer nanocomposites for food packaging applications is highly recommended in most studies [196,197]. Figure 11 shows the schematic development of cellulose-based natural fiber into a biodegradable food packaging material. 

Different uses of AgNPs in the bio-medical field include anti-microbial action, protein recognition, malignancy treatment, clinical textures, anti-microbial catheter, and natural measures [169,170]. This is because AgNPs confer novel properties to nanoparticles, which help with applications in terms of bio-compatibility [174]. AgNPs exhibit nano-items and are commonly used in the form of pieces in therapeutic and other applications including filtration, optics, gadgets, and also in pharmacological treatment processes. Yet, the size and shape characteristics of AgNPs may pose possible threats to human life, and expansive exploration is required to address their combination, depiction, and possible destructiveness [168,169,171]. CNTs are most generally utilized, which comprise graphene layers with sp2 hybridized carbon molecules and covalently reinforcement with three adjoining carbon atoms. The broadest utilization of 3D-printed nano-CNT can be found in hardware. All are explicitly inferable from the extraordinary electrical conductivity of 102 S/m to 107 S/m at 300 K, with most composites experiencing enhanced electrical characteristics.

The utilization of these composite materials incorporates energy-storing devices such as micro super-capacitors and other parts such as transducers, adaptable conductors, resistors, and inductors. As a huge enhancement in electrical conductivity is seen when the concentration of CNTs decreases, 3D-printed composites could be a lightweight, minimal effort, and profoundly successful choice for specific applications. High-level applications that can assimilate the additional expenses include hardware, particularly aviation hardware (which requires lightweight, high-strength, high-temperature-safe composites), and energy (for instance, in nanotube-strengthened elastic seals for enormous oil recuperation stages). Be that as it may, more agreeable interfacial connections may bring about a differential slip system to be used at the interfacial region between the polymer and filler. Subsequently, finding the ideal mix of strengthening and deformational instruments ought to be deliberately measured for the planning of graphene-based PNCs with extreme mechanical properties. Graphene-based PNCs are precisely solid, but also adaptable, which expands the scope of their application areas. 

## 5. Summary and Conclusions

Cellulose and carbon-based nanocomposites are facing a strong challenge in terms of determining optimum synthesis techniques lately. Many new approaches are being continuously developed by researchers to address the involved complexities. The wide application spectrum of carbon- and cellulose-based PNCs makes them into an interesting material for research and motivates the development of these materials on a large scale [191,192]. Parallel efforts for synthesizing cellulose-based PNCs for various applications on large scales may motivate the further development of the PNCs. When taken in terms of the future scope of development, the need for finding an economic way method to synthesize cellulose-based PNCs for various applications must be addressed. Pilot-scale implementation of these nanocomposites is the next step. With these developments, the currently researched synthesis techniques and the materials can be expanded for mass production to cater to the ever-rising demand. Numerous applications of PNCs require high strength, subsequently demanding the expansion of their mechanical properties and elastic moduli along with material homogeneity. Mechanically strong cellulose-based composite membranes are often used in water purification applications and the absorption of toxic gases. The application of cellulose-based composites is also being promoted in wound healing and other biomedical applications. The electrical conductivity of the optoelectronic devices can also be enhanced using chemically modified cellulose-based PNCs. 

This review summarized the ongoing endeavors on the material selection, processing methodologies, handling techniques, hypothetical models, and properties of carbon- and cellulose-based nanoparticle-filled polymer composites. These composites have better characteristics in terms of toughness, but have not exhibited high mechanical characteristics concerning the materials’ elastic moduli and other properties. They have numerous, invaluable advantages, for example, low volume, lighter weight, and imperviousness to temperature changes. These materials lack extensive investigations indicating the primary ideas with respect to their possibilities for mechanical applications. PNCs can be utilized in various applications, for example, biotechnology, nano-electronics, super-capacitors, and bio-sensors. Their usage has been introduced in mechanical applications, including solar cells, radar materials, etc. Subsequently, PNCs are enormously utilized in numerous applications to enhance the electrical, mechanical, and thermal properties of the end products. The interfacial adhesion between nanoparticles and polymeric matrices relies upon the nature of the polymer, as the conceivable cooperation or linkage between the polymer chains significantly influences their properties.

## Figures and Tables

**Figure 1 nanomaterials-13-01803-f001:**
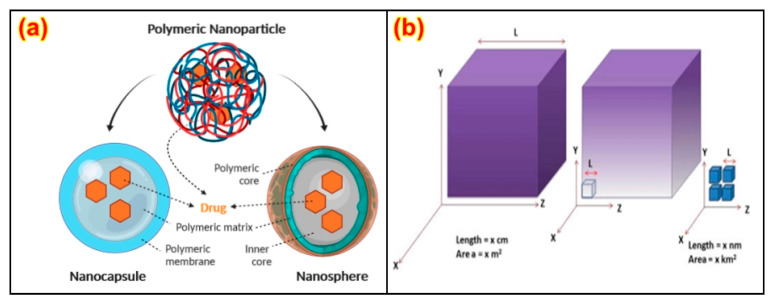
(**a**) Nanoparticles’ structure [28]. Reproduced from Ref. [28] with permission from MDPI. (**b**) surface-to-volume ratio of nanoparticles [29]. Reproduced from Ref. [29] with permission from Springer.

**Figure 3 nanomaterials-13-01803-f003:**
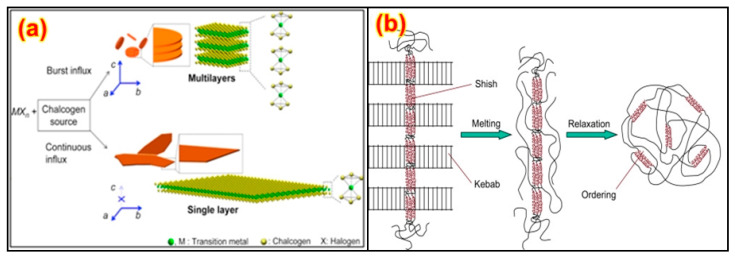
Schematic of (**a**) wet chemical synthesis [95]. Reproduced from Ref. [95] with permission from Elsevier; and (**b**) thermoplastic processing stages [98]. Reproduced from Ref. [98] with permission from Elsevier.

**Figure 4 nanomaterials-13-01803-f004:**
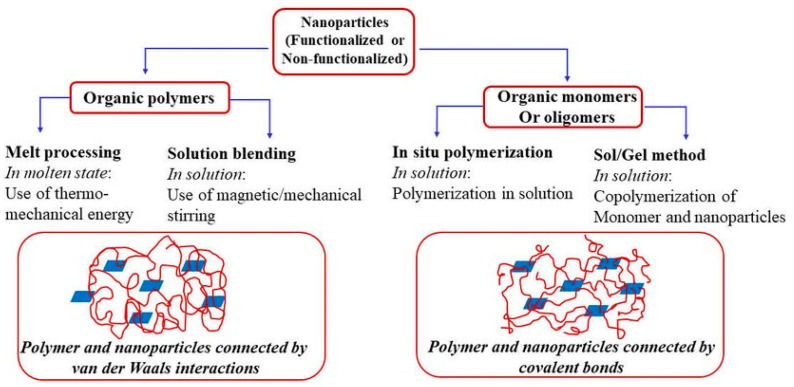
Nanoparticle preparation techniques [123]. Reproduced from Ref. [123] with permission from MDPI.

**Figure 5 nanomaterials-13-01803-f005:**
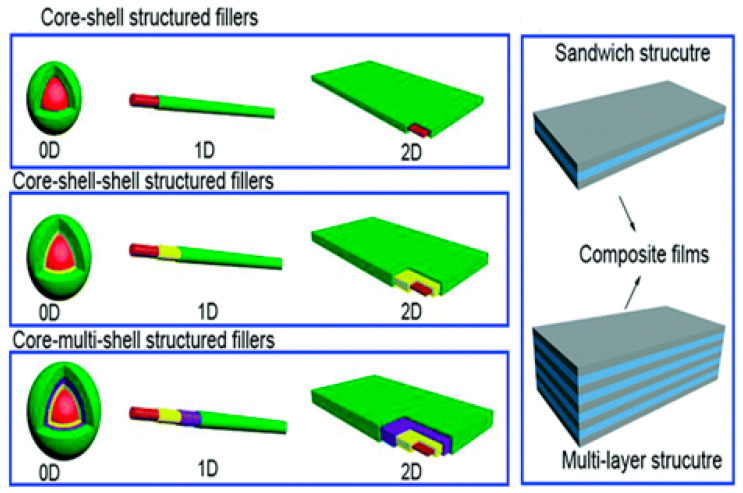
Single and multi-layered nanofillers [82]. Reproduced from Ref. [82] with permission from Royal Society of Chemistry.

**Figure 6 nanomaterials-13-01803-f006:**
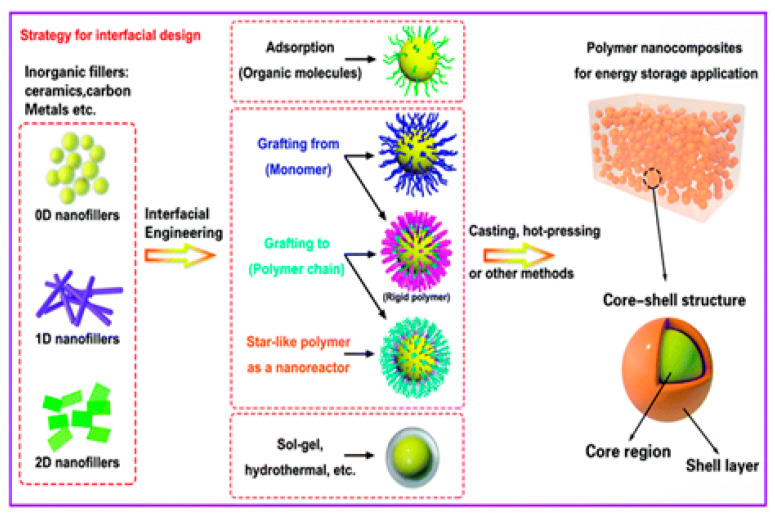
General methods associated with the design of the interface of core–shell fillers [82]. Reproduced from Ref. [82] with permission from Royal Society of Chemistry.

**Figure 7 nanomaterials-13-01803-f007:**
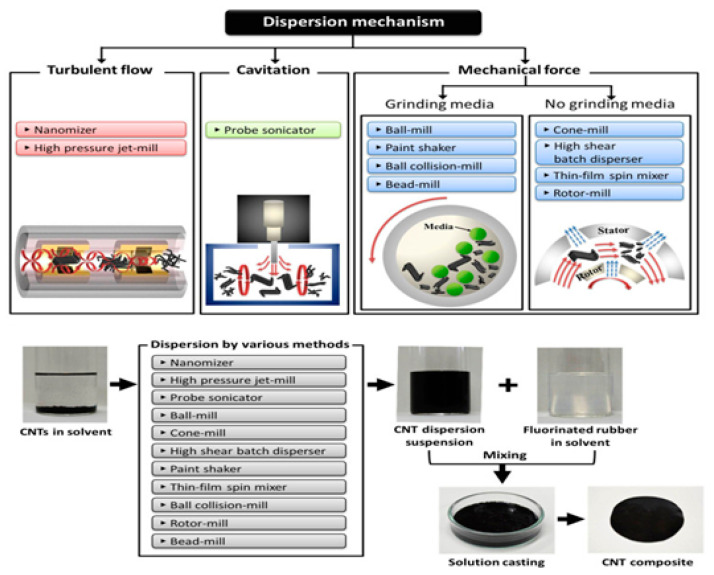
Dispersion techniques [146]. Reproduced from Ref. [146] with permission from Royal Society of Chemistry.

**Figure 8 nanomaterials-13-01803-f008:**
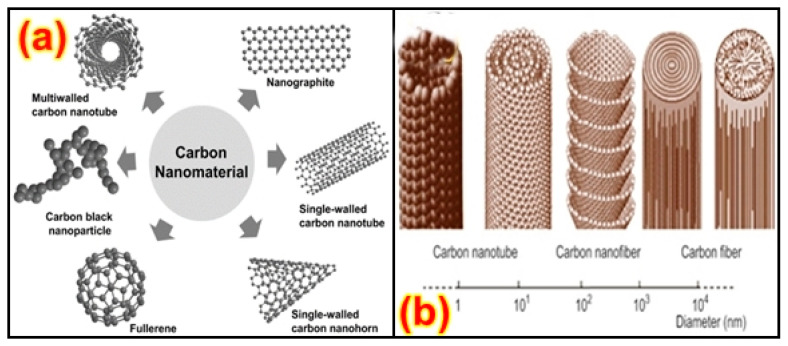
CNT filaments (**a**) Various carbon-based nanomaterials [164]. Reproduced from Ref. [164] with permission from Springer Nature. (**b**) Scale of CNMs [166]. Reproduced from Ref. [166] with permission from MDPI.

**Figure 9 nanomaterials-13-01803-f009:**
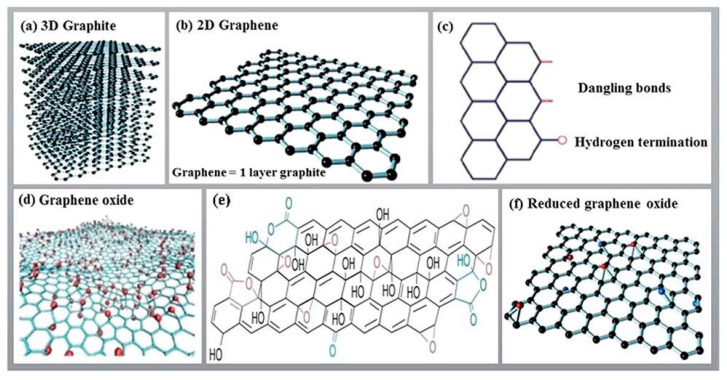
Covalent bonds of (**a**) 3D Graphite, (**b**) 2D Graphene, (**c**) Bonding structure, (**d**) Graphene oxide, (**e**) Hydroxyl bonds arrangement, (**f**) Reduced Graphene Oxide [83]. Reproduced from Ref. [83] with permission from Royal Scoiety of Chemistry.

**Figure 10 nanomaterials-13-01803-f010:**
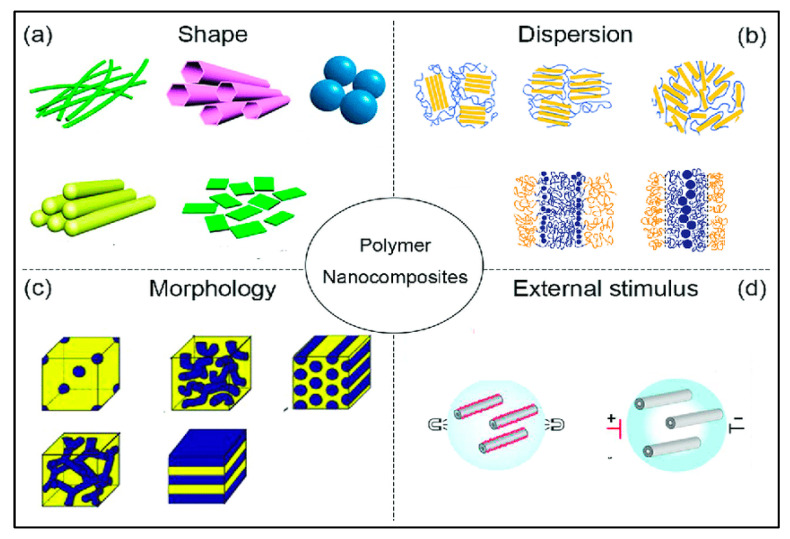
PNC properties: (**a**) shape of the nanopowders; (**b**) dispersion mechanism; (**c**) morphology of the nanopowders; and (**d**) external stimulus used for dispersion [73]. Reproduced from Ref. [73] with permission from MDPI.

**Figure 11 nanomaterials-13-01803-f011:**
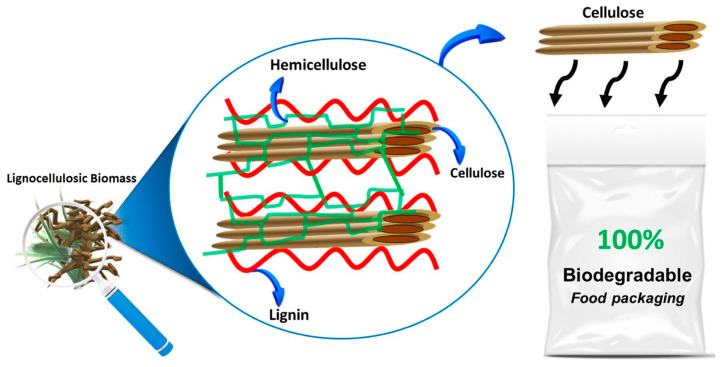
Application of cellulose-based PNCs in food packaging [193]. Reproduced from Ref. [193] with permission from Springer.

**Table 1 nanomaterials-13-01803-t001:** Techniques for the preparation of PNCs.

Thermosetting Polymers	Thermoplastic Polymers
Hand molding and nano-coating	Hot pressing
Inject ram and plunger transfer	Vacuum forming
Closed-mold process	Back pressure method
Plunger and ram-type molding	Cold drawn
Pump nozzle injection	Thermoforming
Vacuum bag molding	Glass fiber thermoplastics
Fiber-reinforced plastics	Drostholm process
Vacuum infusion molding	Injected foam molding

**Table 2 nanomaterials-13-01803-t002:** Mechanical properties of graphene and CNT-based nanomaterials.

Nanocomposites	Nanoparticles Concentration (wt.%)	Fabrication Method	Young’s Modulus (MPa)	Tensile Strength (MPa)	Refs.
Epoxy/graphene	0.62	Sonication	0.8	1.09	[79,82]
Graphene/nano-cellulose	10	In-situ	1.11	1.13	[83]
Polyaniline/graphene	1.6	Solution casting	0.87	1.16	[127]
PMMA/graphene	2–6	Melt blending	0.93	1.21	[82]
Natural rubber/graphene	3.8	In-situ	1.02	2.10	[83,85]
Epoxy/SWCNT	0.5–4	Melt blending	2.04	1.47	[83,84]
PMMA/SWCNT	0.5	Solution casting	1.06	1.9	[128]
Epoxy/MWCNT	0.25–5	In-situ	1.1	1.07	[134]
Polyurethane/SWCNT	0.7	Solution casting	0.6	1.0	[133]
PP/MWCNT	1	Ball milling	1.1	3.7	[140]
PP/SWCNT	0.75	Solution mixing	1.54	1.89	[176]
Polyurethane/SWCNT	0.38	Melt extrusion	1.4	3.6	[139]
HPDE/graphene	3	In-situ	1.05	2.3	[137,138]
LDPE/MWCNT	3–5	Melt mixing	1.56	1.89	[145,146]
Nylon/MWCNT	1	Melt blending	1.06	2.15	[141]
PVA/SWCNT	0.80	Solution casting	1.45	1.92	[139]
PI/SWCNT	0.56	In-situ	1.20	3	[128,133]
PI/MWCNT	1.50	Solution intercalation	1.12	2.47	[142,144]

**Table 3 nanomaterials-13-01803-t003:** Characterization techniques of PNCs.

S. No.	Description	Characterization Technique	Ref.
1.	Confocal laser scanning	Ultrafine microstructure determination	[72,183]
2.	Scanning optical microscopy	Raster scan	[72,100]
3.	Two-photon fluorescence	Biological materials and systems	[100]
4.	Dynamic light scattering technique	Brownian motion and size of the particles	[72]
5.	Brewster angle	Imaging technique for gas liquid interface	[183]
6.	Nano-sight	Characterize nanoparticles from in solution	[100,183]
7.	SEM	Particle shape, size, and morphology	[72]
8.	Electron probe microscopic analysis	Local chemical analysis	[72,100]
9.	Static light scattering	Molecular weight using the relationship between the intensity of light scattered by a molecule	[74,103]
10.	Transmission electron microscopy (TEM)	Particle size and shape and other images at very high resolution	[104,183]
11.	High-resolution TEM	Extensively used to investigate the crystal structures, interfaces, and defects such as dislocations, stacking faults, and grain boundaries of various types of crystalline materials	[183,184]
12.	Low energy electron diffraction technique	Adsorbate bonding of materials	[103,104]
13.	Auger electron	Analysis of chemical surfaces	[100,183]
14.	AFM	Forces between the probe and the sample as an element of their common partition	[100,184]
15.	Magnetic force microscopy	Measures the force of a magnetic field on a tip to elucidate	[103,104]
16.	Scanning tunneling microscopy	Analysis of friction, surface roughness, and surface defects	[99,100]
17.	Atomic probe microscopy	To measure adhesion strength, magnetic forces, and mechanical strengths	[72,100]
18.	Field ion microscopy	To image the arrangement of atoms at the surface	[103,184]
19.	Atomic probe tomography	Nanoscale materials analysis technique that provides 3D spatial imaging and chemical composition measurements with high sensitivity	[104]
20.	Ultraviolet photo-emission	Measurement of kinetic energy spectra of photoelectrons emitted by molecules on the surface	[100]
21.	UV-visible spectroscopy	The absorbance spectra of a compound in solution or as a solid	[72,103]
22.	Atomic absorption spectroscopy	Measuring the concentrations of metallic elements in different materials	[100,104]
23.	Inductively coupled plasma spectroscopy	Technique for trace multi-element and isotopic analysis	[99]
24.	Fluorescence spectroscopy	Analyzes fluorescence from a sample	[100]
25.	Localized surface plasma resonance technique	Analysis of nanoparticle	[103,104]
26.	Rutherford backscattering technique	Elemental analysis using quantitative and qualitative techniques	[100]
27.	Small angle neutron scattering method	Characterization of the surface of the material	[104]
28.	Nuclear reaction analysis method	Thin solid films depth profile	[100,183]
29.	X-Ray diffraction (XRD)	Determine the thickness of thin films and atomic arrangements of amorphous materials	[72]
30.	Raman XRD spectroscopy	Vibration crystal structure analysis	[103,183]
31.	Energy dispersive X-ray spectroscopic method	Elemental constituent analysis	[100]
32.	Small angle X-ray scattering method	Particle sizing in nanoscale	[72,183]
33.	Cathodoluminescence technique	Emission characteristics of materials	[72]
34.	Nuclear magnetic resonance technique	Nuclear species analysis	[100]
35.	Thermo-gravimetric analysis	Relationship between temperature variation and weight loss	[74]
36.	Differential thermal analysis	Reaction heat capacity of the material	[75,183]
37.	Differential scanning calorimetry	Thermal characterization of materials	[90,104]
38.	Nanocalorimetric method	Measures latent fusion heat	[183]
39.	Brunauer-Emmett-Teller technique	Specific areas of nanoparticles, including pore size distribution	[103,104]
40.	Fourier-transform infrared spectroscopy	Obtaining the infrared spectrum of absorption, emission, and the photoconductivity of all types of objects	[100]
41.	Differential thermal analysis	Phase transitions of metals, determining the effect of oxidative or reductive atmospheres on materials	[72,103]
42.	Electron energy loss spectroscopy	Loss of energy, change in momentum, and ionization potential of an atom	[72,183]
43.	Sears technique	Colloidal size	[103]
44.	Laser doppler anemometry	Used for velocity determination and the surface charge of the colloidal particles	[104,105]
45.	Hydrophobic interaction chromatography	Surface hydrophobicity	[100,183]
46.	Pycnometer	The density of nanoparticles is determined	[100]
47.	Gel permeation chromatography	Used to measure the molecular weight of the polymer and its distribution in the matrix	[76,103]

## Data Availability

No associated data is available for this work.

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
