# Peer review of "Carbon and Cellulose-Based Nanoparticle-Reinforced Polymer Nanocomposites: A Critical Review"

_nanomaterials, 2023, doi:10.3390/nano13111803_

Round 1
Reviewer 1 Report
The manuscript entitled “Carbon and cellulose-based nano-particles reinforced polymer nanocomposites: A critical review” has been reviewed. Detailed comments are as follows:
1. The manuscript was not well prepared.
2. Polymer nanocomposites have been extensively studied and reviewed. The title of the present review is about carbon and cellulose-based nanoparticles reinforced polymer nanocomposites. However, carbon and cellulose-based nanoparticles were not well defined, classified and reviewed in the manuscript. On the other hand, the reason of this review was not highlighted in the introduction since there are many books and reviews on the topic of polymer nanocomposites.
3. The definition of nanoparticles is wrong. In Line 35, the authors stated that “nanoparticles width are ranging from 10 to 1000 nm has a novel property ……”. However, according to IUPAC, a nanoparticle is usually defined as a particle of matter that is between 1 and 100 nm in diameter [Vert, M.; Doi, Y.; Hellwich, K. H.; Hess, M.; Hodge, P.; Kubisa, P.; Rinaudo, M.; Schué, F. O. (2012). "Terminology for biorelated polymers and applications (IUPAC Recommendations 2012)". Pure and Applied Chemistry. 84 (2): 377-410.].
4. The structuring of this manuscript should be improved.
5. The figures obtained from other references should be cited in the figure captions.
6. wt. % should be content or concentration.
7. There are many abbreviations in the manuscript. The full names of some abbreviations, especially in the table, were not given. Therefore, an abbreviation list should be added in the manuscript.
8. References should be revised as per guide for authors of Nanomaterials and be checked items by items.
a) The title of some papers, for e.g., Ref. 2, should not be in the form of the capitalization of all first letters.
b) Many volumes and pages or article numbers are missing in some references, for e.g., Ref. 1.
c) Except for the paper in press, DOIs should be removed.
d) Some pages are article numbers (e.g., Ref. 2).
e) The format of book section in some references (e.g., Ref. 4) is not correct.
f) The authors’ first names were not abbreviated, for e.g., Ref. 49.
g) Some volumes are wrong, for e.g., Ref. 88.
Please see Comments and Suggestions for Authors.
Author Response
Comment 1: The manuscript was not well prepared.
Response 1: The suggestion of the reviewer is well-taken. The flow of the manuscript has been suitably altered to enhance the readability and the contents were enriched by citing recently published articles.
Comment 2: Polymer nanocomposites have been extensively studied and reviewed. The title of the present review is about carbon and cellulose-based nanoparticles reinforced polymer nanocomposites. However, carbon and cellulose-based nanoparticles were not well defined, classified and reviewed in the manuscript. On the other hand, the reason of this review was not highlighted in the introduction since there are many books and reviews on the topic of polymer nanocomposites.
Response 2: As per the suggestion of the reviewer, the introduction section has been suitably modified to clearly portray the importance of carbon and cellulose-based nanomaterials.
Comment 3: The definition of nanoparticles is wrong. In Line 35, the authors stated that “nanoparticles width are ranging from 10 to 1000 nm has a novel property ……”. However, according to IUPAC, a nanoparticle is usually defined as a particle of matter that is between 1 and 100 nm in diameter [Vert, M.; Doi, Y.; Hellwich, K. H.; Hess, M.; Hodge, P.; Kubisa, P.; Rinaudo, M.; Schué, F. O. (2012). "Terminology for biorelated polymers and applications (IUPAC Recommendations 2012)". Pure and Applied Chemistry. 84 (2): 377-410.].
Response 3: Authors sincerely apologize for the typographic error. As per the suggestion of the reviewer, the size of nanoparticle has been corrected in the revised manuscript with reference to the suggested article.
Comment 4: The structuring of this manuscript should be improved.
Response 4: As per the suggestion of the reviewer, the manuscript structure has been improved to enhance the readability.
Comment 5: The figures obtained from other references should be cited in the figure captions.
Response 5: As per the suggestion of the reviewer, the figure citations has been included in the figure captions and copyrights for all the images taken from other sources have been obtained.
Comment 6: wt. % should be content or concentration.
Response 6: wt. % is the content of nanoparticles.
Comment 7: There are many abbreviations in the manuscript. The full names of some abbreviations, especially in the table, were not given. Therefore, an abbreviation list should be added in the manuscript.
Response 7: The suggestion of the reviewer is well taken. All the abbreviated terms were given in expanded form during their first usage all through the manuscript.
Comment 8: References should be revised as per guide for authors of Nanomaterials and be checked items by items.
- a) The title of some papers, for e.g., Ref. 2, should not be in the form of the capitalization of all first letters.
- b) Many volumes and pages or article numbers are missing in some references, for e.g., Ref. 1.
- c) Except for the paper in press, DOIs should be removed.
- d) Some pages are article numbers (e.g., Ref. 2).
- e) The format of book section in some references (e.g., Ref. 4) is not correct.
- f) The authors’ first names were not abbreviated, for e.g., Ref. 49.
- g) Some volumes are wrong, for e.g., Ref. 88
Response 8: As per the suggestion of the reviewer, the references were converted into the journal format and the changes were amended in the revised manuscript.
Reviewer 2 Report
In this manuscript (nanomaterials-2404642), the authors reviewed the carbon and cellulose-based nanoparticles reinforced polymer nanocomposites and their applications. The topic of this review has broad appeal, but there are some issues.
1. Introduction: The topic of this review involves carbon materials, but there is a lack of background explanation on carbon materials, such as their characteristics, functions, and types (carbon black, carbon nanotubes, amorphous carbon (daily carbon black), graphene and its derivatives, etc.). In addition, there is not enough discussion about cellulose (cellulose nanocrystals, nano-particles and cellulose paper, etc). It is recommended to fully highlight the writing motivation of this review in the materials.
2. The second (2. Nano-composites) and third parts (3. Polymer nano-composites) should focus on discussing carbon materials and cellulose composite nanomaterials.
3. 4.3 Characterization of PNCs: Is there anything special about characterization for PNCs? Why specifically discuss characterization. The relevant characterizations are only an introduction to concepts and are not discussed with specific examples of PNCs.
4. 5. Applications of PNCs: There are too few application introductions, only two paragraphs. It is recommended to classify and introduce the application, and combine specific examples and figure results.
5. Summary and conclusion: Readers would like to see more constructive strategies. For example, material selection, preparation and application prospects, for example, applications in sensors of carbon and cellulose-based nanoparticles J. Mater. Chem. C, 2023, 11, 5585–5600.
6. References: Check the format of references. The format is incorrect, and some literature information is incomplete.
7. Check English writing and journal format.
Minor editing of English language required.
Author Response
Comment 1: Introduction: The topic of this review involves carbon materials, but there is a lack of background explanation on carbon materials, such as their characteristics, functions, and types (carbon black, carbon nanotubes, amorphous carbon (daily carbon black), graphene and its derivatives, etc.). In addition, there is not enough discussion about cellulose (cellulose nanocrystals, nano-particles and cellulose paper, etc). It is recommended to fully highlight the writing motivation of this review in the materials.
Response 1: As per the suggestion of the reviewer, description regarding carbon and cellulose-based nanomaterials has been included and the importance of this review has been highlighted at the terminal part of the introduction section.
Comment 2: The second (2. Nano-composites) and third parts (3. Polymer nano-composites) should focus on discussing carbon materials and cellulose composite nanomaterials.
Response 2: As per the suggestion of the reviewer, section 2 and 3 has been merged and discussions focusing on carbon and cellulose nanomaterials has been included.
Comment 3: Characterization of PNCs: Is there anything special about characterization for PNCs? Why specifically discuss characterization. The relevant characterizations are only an introduction to concepts and are not discussed with specific examples of PNCs.
Response 3: As per the suggestion of the reviewer, the specific facts about the each characterization techniques has been included beneath each of the techniques apart from their general characteristics.
Comment 4: Applications of PNCs: There are too few application introductions, only two paragraphs. It is recommended to classify and introduce the application, and combine specific examples and figure results.
Response 4: As per the suggestion of the reviewer, the application section of the manuscript has been expanded to cover various applications of the nanocomposites along with some figures.
Comment 5: Summary and conclusion: Readers would like to see more constructive strategies. For example, material selection, preparation and application prospects, for example, applications in sensors of carbon and cellulose-based nanoparticles J. Mater. Chem. C, 2023, 11, 5585–5600.
Response 5: As per the suggestion of the reviewer, summary and conclusion part has been enhanced with the challenges and future perspectives of the PNCs, and the relevant references were added.
Comment 6: References: Check the format of references. The format is incorrect, and some literature information is incomplete.
Response 6: As per the suggestion of the reviewer, the references were formatted according to the journal’s guidelines and complete information of the references are included.
Comment 7: Check English writing and journal format.
Response 7: As per the suggestion of the reviewer, the language throughout the manuscript has been checked and modified. The journal format has been adopted meticulously.
Reviewer 3 Report
The present study reviewed carbon and cellulose-based nano-particles reinforced polymer nanocomposites. However, some minor suggestions are recommended to improve the quality of the present paper.
1) It is recommended to emphasize the novelty of the present study in the introduction section.
2) It is important to disperse conductive fillers in the polymer. There are many different dispersion methods; thus, it is recommended to compare and emphasize the advantages and limitations of the dispersion methods. The following paper (https://doi.org/10.3390/polym14071366) can be used in the present paper.
3) The application of PNC (Section 5) is required to be rewritten. There are many applications using PNC; however, the present study includes only a few applications. Thus, it is recommended to include versatile applications using PNC considering the following papers (https://doi.org/10.1016/j.jmrt.2022.02.134, https://doi.org/10.1016/j.compscitech.2021.108866)
4) The authors are recommended to include some sections regarding the limitations of the present technology and introduce some future works to solve the problems and limitations of the present technology in section 6.
Minor editing of English language required
Author Response
Comment 1: It is recommended to emphasize the novelty of the present study in the introduction section.
Response 1: As per the suggestion of the reviewer, the need of the current review has been included in the introduction section.
Comment 2: It is important to disperse conductive fillers in the polymer. There are many different dispersion methods; thus, it is recommended to compare and emphasize the advantages and limitations of the dispersion methods. The following paper (https://doi.org/10.3390/polym14071366) can be used in the present paper.
Response 2: As per the suggestion of the reviewer, information regarding the conductive fillers has been included in the conventional manufacturing methods section and the reference has been added appropriately.
Comment 3: The application of PNC (Section 5) is required to be rewritten. There are many applications using PNC; however, the present study includes only a few applications. Thus, it is recommended to include versatile applications using PNC considering the following papers (https://doi.org/10.1016/j.jmrt.2022.02.134, https://doi.org/10.1016/j.compscitech.2021.108866)
Response 3: As per the suggestion of the reviewer, the application section of the manuscript has been expanded to cover various applications of the nanocomposites and the relevant references are also included in the list of references.
Comment 4: The authors are recommended to include some sections regarding the limitations of the present technology and introduce some future works to solve the problems and limitations of the present technology in section 6.
Response 4: As per the suggestion of the reviewer, a separate section for challenges and future scope of carbon and cellulose-based PNCs has been included in the revised manuscript.
Round 2
Reviewer 1 Report
The manuscript has been well revised. It can be accepted now.
The manuscript has been well revised. It can be accepted now.
Reviewer 2 Report
Concerns of reviewer have been addressed properly and publication is recommended.
Reviewer 3 Report
The authors have revised the present manuscript considering the reviewer's comments. Thus, the reviewer thinks it can be published in this journal.